# Estimation of Human Casualties Due to Earthquakes: Overview and Application of Literature Models with Emphasis on Occupancy Rate

Vincenzo Manfredi [1],*, Giuseppe Nicodemo [2] and Angelo Masi [1]

[1] School of Engineering, University of Basilicata, 85100 Potenza, Italy; angelo.masi@unibas.it
[2] Ministry of Infrastructures and Transport, 10042 Torino, Italy; giuseppe.nicodemo92@gmail.com
* Correspondence: vincenzo.manfredi@unibas.it

**Abstract:** In the field of seismic risk assessment, the estimation of human casualties is an important task for medical and relief agencies to develop preparedness and emergency management actions. The process of calculating casualties involves several factors along with the associated uncertainties. Despite its complexity and the limited quality and availability of data, many studies have been devoted to this topic in recent decades, but additional effort is required to better analyze these studies by also comparing the results they provide. In the present paper, an extensive literature overview of the main models proposed to estimate casualties is reported. Further, the main factors involved in the available Casualty Estimation Models are also analyzed by analyzing loss scenarios related to two strong Italian earthquakes (1980 Irpinia–Basilicata and 2009 L'Aquila). Comparing estimated vs. collected data, it is found that, in addition to the damage level, both the building material and the occupancy rate at the time of the event significantly impact the estimation of human casualties. As for the occupancy rate, based on the data on the daily life of citizens collected by the Italian Institute of Statistics, the occupancy rate functions for Italian residential building stock are derived and discussed.

**Keywords:** emergency preparedness; seismic consequences; casualty estimation; loss scenarios; occupancy rate function

## 1. Introduction

The main objective of seismic risk assessment for a given area is to calculate the seismic hazard and to convolve it with the vulnerability of the exposed building stock so that damage and losses (human, social and economic) can be predicted [1]. Specifically, the estimation of human casualties (both deaths and injuries) is of great importance for civil protection activities in the management of the emergency after an earthquake. The assessment of casualties is also essential for medical and relief agencies to provide aid and improve their preparedness. However, assessing the number of casualties caused by an earthquake is certainly one of the most complex tasks in the risk assessment process due to the limited availability and quality of information. Moreover, it should be noted that, although casualties are prevailingly related to building damage (especially fully or partially collapsed buildings), as pointed out in past studies (e.g., [2]), an earthquake can cause deaths and injuries in many different ways, among which are secondary hazards (e.g., landslides, mudflows and fires) and large-infrastructure failures.

The type and number of casualties is strictly dependent on the characteristics of the ground shaking and building stock in the stricken area. However, other factors can affect the real impact, such as the different distributions of occupants in the daytime and during different seasons of the year, damage and building collapse mechanisms, and the effectiveness of rescue operations. In view of accounting for the numerous aspects involved and the associated uncertainties, several studies and projects have been devoted in recent

years to the development of methodologies aimed at assessing human casualties (see HAZUS [3], PAGER [4], NERIES [5,6], SYNER-G [7]).

In Italy, the National Risk Assessment (NRA) released in 2018 by the Italian Civil Protection Department (ICPD [8]) provided loss estimations from seismic risk assessment on a national scale. To update the 2018 version of the NRA, the ICPD supported a new research project (2019–2021 ICDP-ReLUIS Project) specifically including the work package WP4 "Seismic Risk Maps-MARS", which, considering the scientific advancements made in the last years, defined a new harmonized seismic risk assessment [9,10]. One of the main objectives was to collect, analyze and update loss models to estimate direct economic losses, unusable buildings and casualties. First, a critical review of the models available in the literature and specific comparisons to better understand their prediction capability and reliability was performed. As for the available models aimed at assessing human casualties, this review showed that there was a need to better analyze them and to compare their results.

Therefore, on the basis of the results obtained from the MARS project, in the present paper, an extensive literature overview of the principal approaches proposed to estimate the expected consequences in terms of casualties (fatalities and injuries) is first reported. After this, the main factors involved in the selected Casualty Estimation Models (CEMs), including damage level, building type and occupancy rate, are analyzed and discussed. Subsequently, comparative analyses among the selected CEMs are performed in terms of loss scenarios based on the data related to two fatal Italian earthquakes, that is, the 1980 Irpinia–Basilicata [11] and the 2009 L'Aquila [12] earthquakes. In order to obtain information related to building characteristics and damage, the dataset of post-earthquake surveys reported on the Da.D.O. (Observed Damage Database) platform [13] is used. The estimated casualties are compared with the actual data, particularly in terms of deaths, and the different factors considered in the CEMs are specifically analyzed in order to investigate their influence on the final results.

Finally, due to the important role of the occupancy rate of buildings, on the basis of the data related to the daily life of citizens collected by the Italian Institute of Statistics [14], functions capable of considering the occupancy rate of residential buildings at the time of the earthquake, specifically calibrated for Italy, are proposed and compared to the available ones.

This paper is organized as follows: after this Introduction, which provides a general background of the research topic of the paper, Section 2 presents a literature overview of the main literature models for estimating human casualties. Section 3 analyses and discusses in detail the factors involved in the available CEMs. Section 4 reports a comparative analysis of CEMs based on the loss scenarios, in terms of death, for the 1980 Irpinia–Basilicata and 2009 L'Aquila earthquakes, thus exploring in depth the role of some significant factors. On the basis of the data provided by the Italian Institute of Statistics [14], Section 5 describes the analyses carried out to derive specific curves of the typical occupancy rates of residential buildings at different times during the scenario event. Finally, Section 6 reports some remarks on the main results obtained in the present paper, as well as on some future developments to be pursued.

## 2. Literature Overview

Large-scale loss estimation methodologies aim to evaluate the seismic consequences related to a large number of assets (buildings, infrastructures, etc.) within a given geographical area (e.g., city or country). Several studies available in the technical literature allow for the estimation of human casualties using different approaches [15], which can be generally classified into (i) empirical, based on the casualty rates of the exposed population as a function of specific factors such as seismic intensity, regional vulnerability level, etc. (e.g., [16,17]); (ii) analytical, based on building damage as an input parameter for evaluating the casualty rates for different structural types and building occupancy (e.g., [2,18]) and (iii) hybrid (e.g., [19]). A wide collection of studies on this topic is reported in the book

"*Human casualties in Earthquakes-Progress in Modelling and Mitigation*" [6] in which both casualty models and new data from studies relating to particular countries or regions are presented. Further, loss estimation tools to be used in real-time and scenario modes are also upgraded (e.g., QLARM [20]).

In the following, some of the main studies and projects related to models for estimating expected casualties are described.

## 2.1. Coburn and Spence Model

This derives from the first model proposed by Coburn and Spence [21], which considers the sum of three different contributions for the estimation of the number of fatalities (K): (i) fatalities due to structural damage (KS); (ii) fatalities due to non-structural damage (K'); and (iii) fatalities due to follow-on hazards (K2). Based on post-earthquake data of the last century from around the world, the authors found that the contribution of KS is dominant (about 75% of the total), particularly for strong shaking levels. Indeed, in stronger events, fatalities are highly affected by structural damage, especially due to fully or partially collapsed buildings, while for smaller earthquakes, non-structural damage represents the main cause [22]. In [2], the same authors proposed an updated version of the "Lethality Ratio" (LR) related to each class of damaged buildings. The LR is defined as the ratio of the number of people killed to the number of occupants present in collapsed buildings and depends on several factors, including building type and function, occupancy rate, the damage/collapse mechanism, ground motion characteristics, occupant behavior and search and rescue (SAR) effectiveness. For each building class, LR is estimated using a set of M-parameters that define (i) the expected proportion of occupants trapped, (ii) the proportion of those trapped who are subsequently rescued and (iii) the distribution of injuries in each group. The number of fatalities is derived from an estimate of the number of collapsed buildings and the LR for each class, as reported in the following equation (adapted from [2]):

$$K_S = D_5 \cdot [M_1 \cdot M_2 \cdot M_3 \cdot (M_4 + (1 - M_4) \cdot M_5)] \tag{1}$$

where:

$K_S$ is the number of fatalities;

$D_5$ is the number of collapsed buildings;

$M_1$ is the average number of people in each collapsed building;

$M_2$ is the percentage of occupants at the time of the earthquake (% of $M_1$ indoors at the onset of ground shaking and influenced by the time of day and use of structures);

$M_3$ is the portion of occupants trapped by the collapse (% of $M_2$ unable to escape);

$M_4$ is the mortality at collapse (% of $M_3$ killed and injured at time zero after collapse);

$M_5$ is the mortality post-collapse (% of the trapped survivors that die before they can be rescued, depending crucially on the effectiveness of SAR).

It is worth specifying that Equation (1) has been modified by the authors from the original version in [2] by introducing the part "$(1 - M_4)$", taking into account that the contribution of the parameter $M_5$ should not include the people already dead at collapse.

Each building class has a specific set of M-parameters depending on the building type, the likely collapse characteristics and the SAR capability. More specifically, $M_3$ is strongly affected by building type and increases with the number of storeys in elevation. Similarly, the portion of occupants killed at collapse ($M_4$) depends on the building type (masonry or reinforced concrete), while $M_5$ (mortality post-collapse) is highly dependent on the effectiveness of SAR, which varies considerably between countries.

The Coburn and Spence model [2] has been widely used in the preparation of loss scenarios in the literature (e.g., [23]).

A further update to the above-mentioned work was proposed by Spence [24] within the LESSLOSS project [18] on the basis of the data obtained from more recent events, such as 1995 Kobe, 1999 Chi-Chi, 1999 Kocaeli, 1994 Northridge and 1999 Athens, and accounting for the rates suggested by ATC-13 [25], Coburn and Spence [21] and FEMA&NIBS [26]. The

classification of M-parameters was modified to include the distribution of fatalities at each damage level and not just those due to the collapse of buildings. Moreover, a classification of injury severity was considered according to the following categories: uninjured (UI), slight injuries (I1), moderate injuries (I2), serious injuries (I3), critical injuries (I4) and deaths (I5). It is assumed that the number of deaths and critical injuries depends primarily on partially or totally collapsed buildings, i.e., D4 and D5 damage levels according to the EMS-98 classification [27], while moderate and serious injuries depend on buildings with lower damage levels.

### 2.2. So and Spence Model

Based on the assembled empirical damage and casualty data in the Cambridge Earthquake Impacts Database (CEQID), So and Spence [28] defined a global casualty estimation model using a semi-empirical approach able to estimate fatality rates as a function of damage level related to building classes.

The total number of fatalities, K, is as follows:

$$K = \sum_l \sum_i [O_{il} * P_{il} * (D5_{il} * L5_i + D4_{il} * L4_i)] \tag{2}$$

where:

$l$ is a town, village or district within which the total population is known or estimated and over which the ground shaking intensity level can be assumed as constant;

$O_{il}$ is the average occupancy rate (i.e., the proportion of the normally resident population who are actually inside the building at the time of the event) in building class i and location l;

$P_{il}$ is the total number of people normally resident in building class i in location l;

$D5_{il}$ and $D4_{il}$ are the proportions of the buildings of class i in location l that are collapsed (damage level D5) and heavily damaged (D4), respectively;

$L5_i$ and $L4_i$ are the lethality rates, i.e., the proportion of occupants killed, in buildings of class i which are collapsed (D5) and heavily damaged (D4), respectively.

The lethality rates are defined for four vulnerability classes (from A to D) related to different types of buildings, as shown in Table 1. The vulnerability classes are similar but not identical to those proposed in the EMS-98 scale [27].

**Table 1.** Lethality rates (L4, L5) at damage levels D4 and D5 for EMS-98 vulnerability classes [28].

| Vulnerability Class | Description of Class | Lethality Rate for D4 (L4) | Lethality Rate for D5 (L5) |
|---|---|---|---|
| A | Weal masonry | 0.05 | 0.200 |
| B | Load-bearing masonry, unreinforced | 0.0195 | 0.078 |
| C | Structural masonry; pre-code reinforced concrete (RC) frame | 0.0625 | 0.250 |
| D * | Moderate code RC frame; concrete shear wall; timber frame | D1: 0.0625 D2:0.0034 | D1: 0.250 D2: 0.013 |
| E | Steel frame; high-code RC | 0.0695 | 0.278 |

* Two subclasses, D1 and D2, were defined because class D contains buildings with widely different lethality potential (RC frame and timber frame).

### 2.3. HAZUS Model

The Federal Emergency Management Agency (FEMA) with the National Institute of Building Sciences (NIBS) developed the HAZUS software based on a standardized earthquake loss estimation methodology. It consists of the following main components: potential Earth science hazards, direct physical damage, induced physical damage and direct economic/social losses. Since the first version [29], numerous updates of the HAZUS software have been made regarding the estimation of earthquake impact, including economic losses and population consequences. The HAZUS model [30] estimates casualties

directly caused by structural or non-structural damage based on four severity levels to classify injuries, from injuries requiring basic medical aid (that could be administered by paraprofessionals, severity level 1) to injuries leading to instantaneous death or mortally (severity level 4).

Casualty rates are obtained by reviewing those suggested in ATC-13 [25] and using limited post-earthquake data, mainly from the USA. The model is based on the studies by Coburn and Spence [21], Murakami [31] and Shiono et al. [32], but, unlike the other approaches, the methodology is in event-tree format. In order to estimate casualties caused by structural damage, the model takes into account a variety of inputs, including damage state probability, building type, occupancy data and event time. The model considers three scenarios based on the time of day: 2:00 a.m. (night-time scenario); 2:00 p.m. (daytime scenario) and 5:00 p.m. (commute time scenario). These scenarios make it possible to estimate the highest casualties for the population at home, at work/school and during rush hour, respectively. The population is distributed into different occupancy categories to consider the fraction of a population component present in an occupancy class for a given time scenario and to divide the population component into indoors and outdoors. Therefore, casualty rates (indoor and outdoor) for each of the 36 building types are defined according to the casualty severity level (from 1 to 4) and damage level (D1—slight, D2—moderate, D3—extensive, D4—complete without collapse, D5—complete with collapse). As an example, Table 2 reports the indoor casualty rates for two building types: concrete moment frames (C1) and unreinforced masonry bearing walls (URM).

**Table 2.** Indoor casualty rates for concrete moment frame structures (C1) and unreinforced masonry bearing wall structures (URM).

| Building Type | Injury Severity | Slight Damage | Moderate Damage | Extensive Damage | Complete Damage |
|---|---|---|---|---|---|
| Concrete moment frame | Level 1 | 0.05 | 0.25 | 1 | 5 *–40 ** |
| | Level 2 | - | 0.03 | 0.1 | 1 *–20 ** |
| | Level 3 | - | - | 0.001 | 0.01 *–5 ** |
| | Level 4 | - | - | 0.001 | 0.01 *–10 ** |
| Unreinforced masonry bearing walls | Level 1 | 0.05 | 0.35 | 2 | 10 *–40 ** |
| | Level 2 | - | 0.04 | 0.2 | 2 *–20 ** |
| | Level 3 | - | 0.001 | 0.002 | 0.02 *–5 ** |
| | Level 4 | - | 0.001 | 0.002 | 0.02 *–10 ** |

* Values related to partial collapse of buildings; ** values related to total collapse of buildings.

### 2.4. Jaiswal and Wald Model

In the framework of the "Prompt Assessment of Global Earthquakes for Response (PAGER)" project developed by the U.S. Geological Survey (USGS), Jaiswal and Wald [17] proposed an empirical approach to estimating the number of deaths. More specifically, based on a global catalogue of all significant earthquakes in the world (1973–2007), a set of empirical vulnerability relationships was derived to provide a fatality rates depending on the Modified Mercalli Intensity (MMI) values. The fatality rate is defined as the ratio of the total number of deaths to the total population exposed at each level of shaking intensity. The expected number of deaths Ei can be evaluated using the following equations:

$$E_i = \sum_j v_i(S_j) P_i(S_j) \tag{3}$$

$$v(S_j) = \Phi\left[\frac{1}{\beta} \ln\left(\frac{S_j}{\theta}\right)\right] \tag{4}$$

where:

$S_j$ is a set of discrete values of shaking intensity at level j (e.g., 5.0; 5.5 of MMI);
$v_i(S_j)$ is the fatality rate as a function of shaking intensity S at level j for event i;
$P_i(S_j)$ is the estimated population exposed to shaking intensity S at level j for event i;
$\Phi$ is the standard normal cumulative distribution function;

θ is the mean of the natural logarithm of the intensity measure;

β is the standard deviation of the natural logarithm of the intensity measure.

As can be seen in Equations (3) and (4), the fatality rates are expressed in terms of a two-parameter lognormal cumulative distribution of shaking intensity (β and θ). The population exposed at each level of shaking intensity is determined by overlaying the USGS ShakeMap with the LandScan global population maps [33]. Where casualty data related to historical earthquakes were available, the parameters of the distribution were obtained by minimizing the residual error (estimated vs. recorded deaths) evaluated as the square error (L2 norm), the log-residual error (G norm) or their combination (L2G norm). For simplicity, only the expression to evaluate L2G norm is reported:

$$\varepsilon_{3,k} = \text{L2G norm} = \ln\left(\sqrt{\frac{1}{N}\sum_{i=1}^{N}(E_i - O_i)^2}\right) + \sqrt{\frac{1}{N}\sum_{i=1}^{N}\left[\ln(E_i - O_i)^2\right]} \tag{5}$$

where:

k indicates the country or a geographic location;

N represents the historical fatal earthquakes in the k region;

$O_i$ is the number of recorded deaths for an i earthquake.

Where empirical data were lacking, based on specific indicators able to associate countries with similar vulnerability, regionalization schemes aimed at aggregating fatal events were proposed. The schemes combine information related to geography, climatic similarities, building inventory and socioeconomic indicators (see [34]).

### 2.5. Zuccaro and Cacace Model

Starting with the study by Coburn and Spence [21], Zuccaro and Cacace [35] presented a model for evaluating seismic casualties in Italy. Specifically, the model is based on the assessment of four fundamental parameters: casualty percentage by building type and damage level; mean number of inhabitants by building type; occupancy rate by hour of the day and week; and touristic index by town and period of the year.

Based on these parameters, the number of deaths ($N_d$) and injuries ($N_i$) is determined using the following expressions:

$$N_d = TI_c \sum_{t=1}^{4}\sum_{j=1}^{5} N_{t,j} NO_t QD_{t,j} \tag{6}$$

$$N_i = TI_c \sum_{t=1}^{4}\sum_{j=1}^{5} N_{t,j} NO_t QI_{t,j} \tag{7}$$

where:

t is the building type (t = 1, . . ., 4);

j is the damage level (j = 1, . . ., 5);

$N_{t,j}$ is the number of buildings of type t w damage level j;

$NO_t$ is the number of occupants (at the time of the event) by building type;

$TI_c$ is the touristic index by city;

$QD_{t,j}$ ($QI_{t,j}$) is the proportion of deaths (injuries) by building type and damage level.

The authors assumed that the number of deaths and injuries is strictly correlated with structural damage, and that the probability of the injury or death of building occupants depends only on the EMS-98 damage levels D4 (i.e., very heavy damage) and D5 (i.e., destruction). Based on the observations of past events, the model also considers the structural type, i.e., reinforced concrete (RC) and masonry. Table 3 shows casualty percentages (as a proportion of the building occupants) according to damage level, structure type (in terms of "vertical structures") and EMS-98 vulnerability classes (A, B, C, D).

As for the number of occupants, this depends on the volume, type and age of the building. In general, due to the lack of information about volume, the number of occupants is estimated considering the total population and building types within the studied area. This calculation can vary considerably for buildings with different use classes (e.g.,

residential, school or industrial). Since the number of casualties depends on the overall population density at the time that the event occurs, the authors suggest taking into account the mobility (i.e., from the satellite towns towards the larger urban settlements) and the variation in population during a month (long period), a day of the week and an hour (short period), although no specific function was provided in order to calculate them.

**Table 3.** Casualty percentages by damage level and building type [35].

| Casualty Percentage | Damage Level D4 | Damage Level D5 | Vertical Structure | Vulnerability Class |
|---|---|---|---|---|
| QD | 0.04 | 0.15 | Masonry | A or B or C |
| QD | 0.08 | 0.3 | RC | C or D |
| QI | 0.14 | 0.7 | Masonry | A or B or C |
| QI | 0.12 | 0.5 | RC | C or D |

*2.6. SYNER-G Model*

As part of the SYNER-G project [36], a model was developed for estimating the number of casualties among building occupants at the time of an earthquake. The proposed methodology considers several elements, such as building occupancy, damage probability matrices, seismic intensity, building/casualty type and casualty/damage ratios. These elements are combined according to the following expression:

$$N_d = \sum_{t=1}^{3} \sum_{d=1}^{6} \sum_{i=1}^{4} N_{t,d,i} CR_{t,d,i} NO_t \tag{8}$$

where:

$N_d$ is the number of deaths;

t is the building/casualty type;

d is the damage level (from D0 to D5);

i is the seismic intensity level;

$N_{t,d,i}$ is the number of buildings of type t with damage level d at seismic intensity level i;

$CR_{t,d,i}$ is the proportion of deaths by building type, damage level and seismic intensity;

$NO_t$ is the number of occupants (at the time of the event) by building type t.

On the basis of data from three Italian earthquakes (1980 Irpinia; 1976 Friuli; and 2009 L'Aquila), three "superclasses" of building types are defined in terms of capability to produce casualties: 1-BC related to reinforced concrete buildings (very high casualty potential); 2-BC related to stone, brick or block masonry walls with reinforced concrete floors/roofs (high casualty potential); and 3-BC related to stone, brick or block masonry walls with timber rubble masonry, timber or steel joist floors/roofs (moderate casualty potential).

The casualty ratio (CR) is defined as the ratio of the number of people killed to the number of occupants present in damaged buildings of a given class. Considering the data in the global CATDAT damaging earthquakes database [37] for the three Italian earthquakes, the semi-empirical CRs for each degree of macroseismic intensity (from 6 to 9 on the EMS-98 intensity scale), building-casualty type and damage level were defined (see Table 4). It is worth noting that CRs are also defined using the half degree system of macroseismic intensity (i.e., 6, 6.5, etc.).

As regards the estimation of the number of occupants, the temporal occupancy model of Coburn and Spence [2] was used to determine the population in different building types at the time of the earthquake.

**Table 4.** Casualty ratios according to the SYNER-G model [36].

| Intensity 6 | D0 | D1 | D2 | D3 | D4 | D5 |
|---|---|---|---|---|---|---|
| 1-BC | 0 | 0 | 0 | 0.0011 | 0.0027 | 0.0067 |
| 2-BC | 0 | 0 | 0 | 0.0005 | 0.0013 | 0.0033 |
| 3-BC | 0 | 0 | 0 | 0 | 0.007 | 0.0017 |
| **Intensity 7** | **D0** | **D1** | **D2** | **D3** | **D4** | **D5** |
| 1-BC | 0 | 0 | 0.009 | 0.0021 | 0.0053 | 0.0133 |
| 2-BC | 0 | 0 | 0 | 0.0011 | 0.0027 | 0.0067 |
| 3-BC | 0 | 0 | 0 | 0.0005 | 0.0013 | 0.0033 |
| **Intensity 8** | **D0** | **D1** | **D2** | **D3** | **D4** | **D5** |
| 1-BC | 0 | 0.0009 | 0.0021 | 0.0053 | 0.0133 | 0.0333 |
| 2-BC | 0 | 0 | 0.0011 | 0.0027 | 0.0067 | 0.0167 |
| 3-BC | 0 | 0 | 0.0005 | 0.0013 | 0.0033 | 0.0083 |
| **Intensity 9** | **D0** | **D1** | **D2** | **D3** | **D4** | **D5** |
| 1-BC | 0 | 0.0048 | 0.0073 | 0.0182 | 0.0454 | 0.1136 |
| 2-BC | 0 | 0.0024 | 0.0036 | 0.091 | 0.0227 | 0.0568 |
| 3-BC | 0 | 0.002 | 0.003 | 0.0076 | 0.0189 | 0.0473 |

### 2.7. Italian NRA Model

In the 2018 "Italian National Risk Assessment (NRA)" [8], released by the Italian Civil Protection Department (ICPD), the results in terms of deaths and injuries were evaluated based on a specific model described in [38] which, in turn, refers to the results reported in [39,40].

In these studies, based on statistical correlations between damage and effects due to Italian earthquakes, the number of fatalities and severely injured residents was assumed to be 30% of the population living in collapsed buildings. This value was also adopted in [41].

In the 2018 NRA model [38], the ratio of injuries ($N_i$) or deaths ($N_d$) is determined for the EMS-98 D4 and D5 damage levels (the most severe ones) as follows:

$$N_d = \sum_{j=1}^{n_M} \left[ \left( O_{Mj,D4} \cdot p_{d,D4} + O_{Mj,D5} \cdot p_{d,D5} \right) \right] + \sum_{l=1}^{n_{RC}} \left[ \left( O_{RCl,D4} \cdot p_{d,D4} + O_{RCl,D5} \cdot p_{d,D5} \right) \right] \quad (9)$$

$$N_i = \sum_{j=1}^{n_M} \left[ \left( O_{Mj,D4} \cdot p_{i,D4} + O_{Mj,D5} \cdot p_{i,D5} \right) \right] + \sum_{l=1}^{n_{RC}} \left[ \left( O_{RCl,D4} \cdot p_{i,D4} + O_{RCl,D5} \cdot p_{i,D5} \right) \right] \quad (10)$$

where:

$n_M$ and $n_{RC}$ are the number of URM and RC building types, respectively;

$O_{Mj,D4/D5}$ ($O_{RCl,D4/D5}$) is the number of occupants in URM (RC) buildings who experienced damage levels of D4 and D5;

$p_{d,D4}$, $p_{d,D5}$ ($p_{i,D4}$, $p_{i,D5}$) are the percentages of deaths (injured) compared to the occupants in buildings with damage levels of D4 and D5;

It is worth noting that, in the NRA model [38], the percentages for dead and injured people are irrespective of building type, i.e., RC and masonry, as reported in Table 5.

**Table 5.** Casualty percentage for computation of human losses in the NRA model [38].

| Expected Casualties | Damage Level D4 | Damage Level D5 |
|---|---|---|
| Dead $p_d$ (%) | 1 | 10 |
| Injured $p_i$ (%) | 5 | 30 |

### 3. Analysis of Casualty Estimation Models in Terms of Factors Involved

According to the Casualty Estimation Models (CEMs) available in the literature, many factors can affect the real impact of an earthquake in terms of casualties. For the sake of clarity, Table 6 summarizes the main factors and related criteria considered in the CEMs described above, in terms of:

- Damage levels;
- Building classification in terms of structural types or vulnerability classes;
- Earthquake intensity in terms of macroseismic values;
- Factors explicitly determining the fatality rate (DL = damage level; BC = building classification; MI = macroseismic intensity);
- Explicit consideration of occupancy rate at the time of the event;
- Explicit consideration of search and rescue (SAR) capability;
- Injury severity classification;
- Data source and related application context.

**Table 6.** Summary of the factors involved in the Casualty Estimation Models (CEMs) described in Section 2.

| CEM | Damage Level | Building Classification | Earthquake Intensity | Fatality Rate | Occupancy Rate | SAR Capability | Injury Classification | Data Source |
|---|---|---|---|---|---|---|---|---|
| Coburn and Spence [2] | D5 (Collapse) | URM-RC | Yes | DL; BC; MI | Yes | Yes | 4 categories (From death to light injury not necessitating hospitalization) | Global |
| So and Spence [28] | D4–D5 (EMS-98) | Vc = A, B, C, D1, D2, E | Yes | DL; BC | Yes | - | Dead | Global |
| HAZUS [30] | From slight damage to collapse | 36 types | - | DL; BC | Yes | - | 4 categories (From instantaneous death to injuries requiring basic medical aid) | USA |
| Zuccaro and Cacace [35] | D4–D5 (EMS-98) | URM-RC or Vc = A, B, C, D | - | DL; BC | Yes | - | Dead and injured | Italy |
| SYNER-G [36] | D1–D5 (EMS-98) | Superclass category (1-BC, 2-BC, 3-BC) | Yes | DL; BC; MI | Yes | - | Dead | Italy |
| Italian NRA [38] | D4–D5 (EMS-98) | URM-RC | - | DL | - | - | Dead and injured | Italy |

It is worth noting that the CEM by Jaiswal and Wald [17] is not reported in Table 6. In fact, unlike the other CEMs, the fatality rate obtained using the Jaiswal and Wald model [17] is expressed as a proportion of the population exposed at each intensity level and depends on the shaking intensity according to a two-parameter lognormal cumulative distribution function.

As can be seen in Table 6, all models are based on building damage, albeit with different numbers of considered levels. Most CEMs attribute the casualties only to buildings damaged at level D5 (collapse) and/or D4 (heavy damage) according to the EMS-98 classification. On the contrary, the CEMs proposed in [30,36] also attribute the casualties to buildings with damage levels lower than D4. However, it is worth pointing out that previous studies (e.g., [22]) indicated that casualties associated with lower damage levels are a small percentage of the total.

The classification of building stock is defined in terms of structural types (e.g., [2]) or vulnerability classes (e.g., [28]). Although structural types and vulnerability classes generally represent a building's susceptibility to damage due to a given intensity of ground shaking, the aim of the classification is to distinguish building types to account for their potential to produce casualties (see, e.g., "superclasses" in SYNER-G project). In other words, different building types, with the same damage level, can have different mechanisms of damage and collapse, and this can affect the expected number of casualties.

As can be seen in Table 6, two models, i.e., Coburn and Spence [2] and SYNER-G [36], explicitly take into account the role of earthquake intensity (in terms of macroseismic values) in fatality rates, while the other CEMs consider it in the derivation but not explicitly in fatality rates (e.g., [28]).

As for fatality rates, the CEMs are explicitly based on one or more of three factors: damage level (DL); building classification (BC); and macroseismic intensity (MI). The fatality rate according to the 2018 NRA model [38] is evaluated only as a function of the damage levels D4 and D5.

Estimating the number of people per building and the occupancy at the time of the earthquake are two of the main sources of uncertainty in CEMs. All the CEMs in Table 6 explicitly consider the occupancy rate at the time of the event, except for the CEM adopted in the NRA [8], which assumes that the occupancy rate is implicitly considered in the fatality rate values. As extensively reported in Section 5, only Coburn and Spence [2] provide the specific distribution of the population during the day for both urban and rural areas. Similarly, only the Coburn and Spence model [2] considers the effectiveness of post-event search and rescue (SAR) activities, while the other authors neglect this factor, mainly due to its poor reliability and applicability.

Generally, the considered CEMs make it possible to evaluate the expected number of both deaths and injuries. Two CEMs, i.e., [2,21], provide a classification in terms of injury severity, i.e., from dead to slightly injured, which is very useful for medical and relief agencies to enhance preparedness measures and for the engineering community to improve emergency planning.

As can be seen in Table 6, the CEMs described above were derived from different data sources: some were derived considering information related to a specific country (e.g., Italy for Zuccaro and Cacace [35]), while other models were based on information related to a wider context (e.g., Coburn and Spence [2]). Therefore, additional caution should be exercised when models derived from a specific area are used for different and/or larger areas. As an example, the possible limitations and uncertainties in applying the HAZUS model [30], which is based mainly on American earthquake data, in a different context, should be considered in the results.

Further limitations related to the considered CEMs derive from the fact that casualties occurring outside buildings are neglected. Therefore, the number of casualties deriving from secondary hazards (such as landslides, fires and tsunamis), infrastructure failures (viaducts or bridges), medical causes (panic or heart attacks) and many other causes is not included.

Finally, in addition to the numerous uncertainties directly associated with the factors involved in the CEMs, an important source of uncertainty is the poor quality and availability of data. Casualty data from past earthquakes is often inconsistent and unreliable, mainly due to the lack of standard methodologies for reporting useful information (e.g., type and cause of casualty, building type, etc.). This lack often leads to an estimation of factors based on engineering aspects and not on empirical data. As a result of these large uncertainties, CEMs should provide a range of confidence, which is extremely important in the decision-making processes of local and national authorities as it enables them to be aware of errors.

## 4. Comparison of Casualty Estimation Models Based on Loss Scenarios

In order to discuss possible differences among the considered CEMs and evaluate the factors that most influence the estimation of human casualties, in this section, some comparative analyses are performed in terms of loss scenarios related to two Italian earthquakes, that is, the 1980 Irpinia–Basilicata [11] and the 2009 L'Aquila earthquakes [12]. These two fatal events were selected primarily for the accuracy and reliability of the recorded data. The comparison was carried out in terms of the number of deaths since all the CEMs described above provide this number, and considering that the actual data on deaths are more reliable than the one related to injuries.

In order to obtain information related to building types and damage, the dataset of post-earthquake surveys reported on the Da.D.O. (Observed Damage Database) platform [13] was used. It is worth noting that the HAZUS model [30], based on American earthquake data, was not considered because it would be misleading to apply it to the Italian built environment.

Data for the 1980 Irpinia–Basilicata earthquake refer to 41 municipalities struck by the earthquake that occurred on 23 November at 19:34:52 local time with Mw equal to 6.81. The macroseismic intensity values according to the MCS scale were obtained from the Italian Macroseismic Database (DBMI15, also reported in the Da.D.O. database).

For these 41 municipalities, for which the macroseismic intensity values are in the range of V-X MCS, a building-by-building survey of the damage and structural characteristics is available, as reported in [11]. According to the survey, the total number of buildings is equal to 38,079, while the total number of inhabitants, calculated from 1971 ISTAT data, is equal to about 140,000.

The number of actual deaths was obtained from the following sources:

- Cambridge Earthquake Impact Database [42];
- Catalogue of strong earthquakes in Italy (461 B.C.–1997) and Mediterranean area (760 B.C.–1500) (CFTI5Med [43]) created by the National Geophysics and Volcanology Institute (INGV);
- Geological and geo-environmental instability in Italy from the post-war period to 1990 [44].

Data for the 2009 L'Aquila earthquake refer to the urban area of L'Aquila hit by the 2009 L'Aquila earthquake, which occurred on 6 April at 03:32:39 local time with a Mw equal to 6.3. Macroseismic intensity equal to VIII-IX in terms of both MCS and EMS-98 scales was assigned to the considered area [12,45].

The dataset consists of about 11,300 buildings and about 40,000 inhabitants (extrapolated from ISTAT data). It is worth noting that only the urban area of L'Aquila was selected due to issues regarding the database's completeness of post-earthquake surveys (e.g., [46]).

*4.1. Application of CEMs: Comparison of Actual vs. Estimated Casualties*

As a first step, the distribution of the building stock in terms of structural types was determined and is reported, respectively, in Table 7a for the 41 municipalities affected by the 1980 Irpinia–Basilicata earthquake (hereafter DB Irpinia–Basilicata), and Table 7b for the L'Aquila urban area data related to the 2009 L'Aquila earthquake (hereafter DB L'Aquila).

The results show that most of the buildings included in the DB Irpinia–Basilicata have a masonry structure (about 79%), while only 10% were built with a reinforced concrete (RC) structure. As for the L'Aquila urban area, about 43% of the buildings have a masonry structure, and a similar percentage (40%) has an RC structure. Due to the prevalence of these two building types, the following comparative analyses were carried out by considering only the masonry and RC buildings.

By applying the criteria defined by Chiauzzi et al. [47] and Dolce et al. [48], seismic vulnerability according to the four classes, i.e., "A", "B", "C", and "D", defined using the EMS-98 scale [27], was assigned to each surveyed building. In this context, it is worth noting that the EMS-98 vulnerability class was assigned on the basis of structural characteristics or construction age. Specifically, the vulnerability of masonry buildings was assigned on the basis of both horizontal and vertical structural type. As for RC buildings, vulnerability classes were assigned on the basis of the construction age: medium–low vulnerability (i.e., "C") was assigned to structures without an earthquake-resistant design (i.e., built before 1980), while the lowest-vulnerability (i.e., "D") class was assigned to buildings designed according to modern anti-seismic criteria (i.e., built or retrofitted after 1980). As a result, the DB L'Aquila shows that about 70% of the building stock is characterized by the low and medium–low vulnerability classes. As for the Irpinia–Basilicata database, about 57% of the building stock belongs to class "A" (i.e., high vulnerability), while about 20% belongs to class "C" (i.e., medium–low vulnerability).

**Table 7.** Distribution of the building stock related to the DB Irpinia–Basilicata (a) and DB L'Aquila (b) in terms of building types.

| (a) | | |
|---|---|---|
| **DB Irpinia–Basilicata** | | |
| **Building Type** | **Building Number** | **Percentage** |
| Masonry | 30,033 | 79% |
| RC | 4185 | 11% |
| Steel | - | - |
| Mixed (RC-Masonry) | 3666 | 9% |
| Undefined | 195 | 1% |
| Total | 38,079 | |
| **(b)** | | |
| **DB L'Aquila** | | |
| **Building type** | **Building Number** | **Percentage** |
| Masonry | 4847 | 43% |
| RC | 4486 | 40% |
| Steel | 274 | 2% |
| Mixed (RC-Masonry) | 774 | 7% |
| Undefined | 930 | 8% |
| Total | 11,311 | |

Since damage levels in the considered CEMs are presented in terms of EMS-98 classification, while data related to the two databases were collected through different approaches, i.e., the Irpinia 1980 form [13] and the AeDES 06/2008 form [49], respectively, for the Irpinia–Basilicata and L'Aquila earthquakes, the conversion schemes proposed in Dolce et al. [13] were applied (with reference to only vertical structure damage). Figure 1a,b show the distribution in terms of EMS-98 damage levels for the DB Irpinia–Basilicata and the DB L'Aquila, respectively. As can be seen, about 10% of the buildings in the DB Irpinia–Basilicata were heavily damaged (i.e., D4) and collapsed (i.e., D5). For the DB L'Aquila, about 12% of the buildings suffered a damage level $\geq 4$.

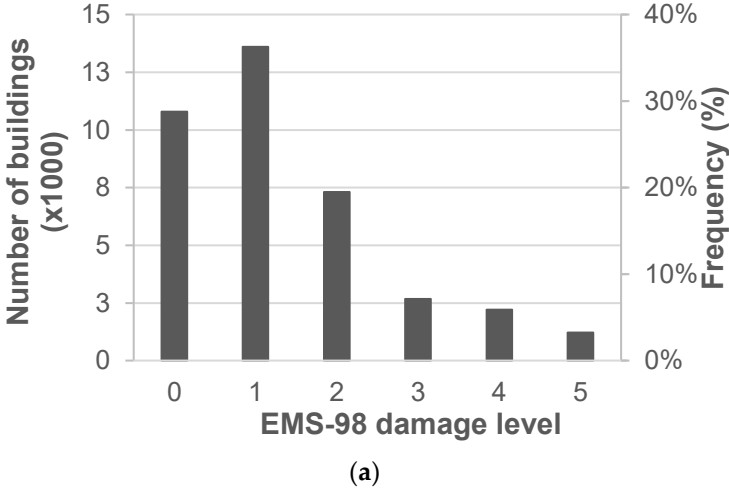

(a)

**Figure 1.** *Cont.*

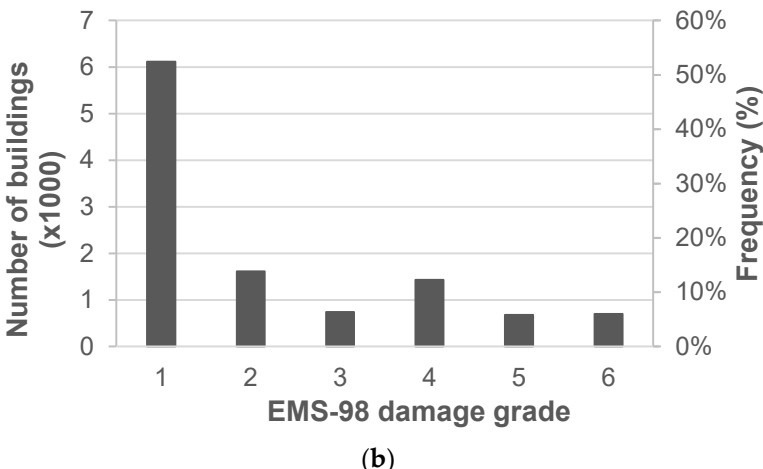

(**b**)

**Figure 1.** Damage distribution in terms of EMS-98 levels for the Irpinia–Basilicata (**a**) and L'Aquila (**b**) scenario.

In order to estimate the number of inhabitants in the different building types, the distribution in terms of number of storeys for reinforced concrete (RC) and masonry (URM) buildings was obtained from the Da.D.O. dataset. Consequently, the mean number of people by building type (URM or RC) was estimated according to the following expression:

$$n°_{MED}\frac{people}{building\_M/RC} = n°_{MED}\frac{people}{storey} \cdot n°_{MED}\frac{storeys}{building_{M/RC}} \tag{11}$$

where:

$n°_{MED}\frac{people}{building\_M/RC}$ is the mean number of people by building type (URM or RC);

$n°_{MED}\frac{people}{storey}$ is the mean number of people in each storey;

$n°_{MED}\frac{storeys}{building_{M/RC}}$ is the mean number of storeys for single URM and RC buildings.

As a result, for the DB L'Aquila, the mean number of people by URM and RC building type is equal to 3 and 4.5, respectively. For the DB Irpinia–Basilicata, values equal to 3.5 and 4.5 for the URM and RC types were estimated, respectively.

The Coburn and Spence model [2], the only one that provides the distribution of population at different times of the day, was adopted to estimate the occupancy rate at the time of the earthquake (see Figure 2) for all CEMs. More specifically, by considering the curve related to the residential buildings (by urban population) and the time of the earthquake (i.e., 19:34:52 local time for the 1980 Irpinia–Basilicata earthquake, and 03:32:39 local time for 2009 L'Aquila earthquake), the values of occupancy rates are equal to about 67% and 72%, respectively.

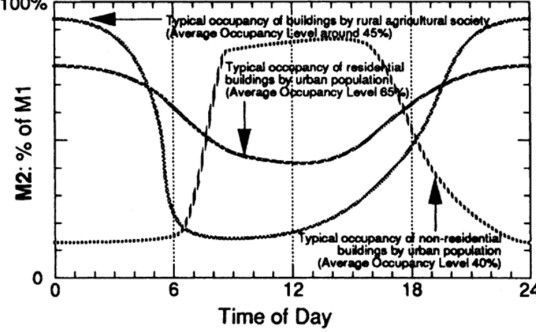

**Figure 2.** Occupancy at the time of the earthquake (from [2]).

Finally, the CEMs were applied and the numbers of deaths were evaluated and are shown in Figure 3, along with a comparison with the actual number of deaths (black line). For the sake of simplicity, in Figure 3 the CEMs are indicated with an acronym, also used in the following text.

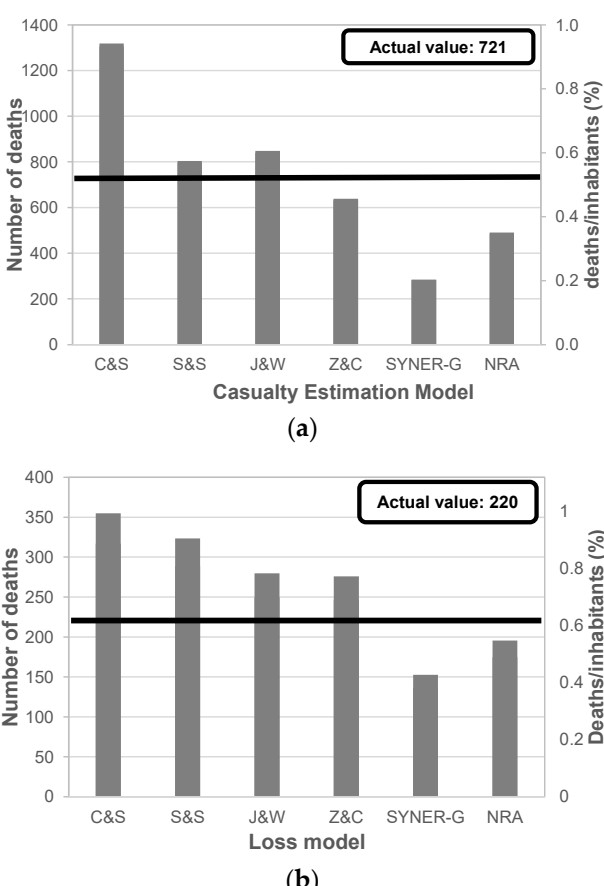

**Figure 3.** Comparison among the CEMs applied to the Irpinia–Basilicata (**a**) and L'Aquila (**b**) scenarios. C&S—Coburn and Spence model [2]; S&S—So and Spence model [28]; J&W—Jaiswal and Wald model [17]; Z&C—Zuccaro and Cacace model [35]; SYNER-G—SYNER-G model [36]; NRA—Italian NRA model [38].

It is worth highlighting that, according to the values suggested for Italy by Jaiswal and Wald [2], in the J&W model, the parameters (θ and β) are set to 13.23 and 0.18, respectively. Similarly, in the C&S model [2], the values of the M5 parameter adopted for the L'Aquila earthquake derive from the situation "Community + emergency squads + SAR experts after 36 h" reported in [2], for which the suggested values are 0.45 for URM buildings and 0.7 for RC ones. As for the Irpinia–Basilicata earthquake, due to the poor SAR capability experienced during the event, the assumed values (i.e., 0.95 for URM buildings and 1 for RC ones) derive from those suggested by Coburn and Spence in [2] for the worst situation (i.e., "Community incapacitated by high casualty rate").

First of all, Figure 3 shows high dispersion between the estimated values with the CEMs, particularly for the Irpinia–Basilicata scenario (Figure 3a), mainly due to the limited reliability of the data available on the Da.D.O. platform. For both scenarios, C&S and SYNER-G provide, respectively, the highest number of deaths (about 355 for the L'Aquila scenario and about 1318 for the Irpinia–Basilicata scenario) and the lowest ones (about 153 for the L'Aquila scenario and about 290 for the Irpinia–Basilicata scenario).

With regard to the actual number of deaths related to the Irpinia–Basilicata earthquake (equal to 721), C&S, S&S and J&W provide higher values, while Z&C, SYNER-G and the

NRA underestimate the number of casualties. However, the results from S&S and Z&C appear to be closer to the actual data.

A similar trend is also found in the case of the L'Aquila scenario, except for Z&C, which provides an overestimation of the actual value.

The observed differences derive from various factors involved in the CEMs, and their influence on the final results is very difficult to assess. As an example, the consideration of SAR effectiveness in C&S is certainly one of the causes of the higher values compared to other CEMs, as can be seen particularly for the 1980 earthquake when the SAR effectiveness was very poor ("*There was not the immediate relief that there should have been. Still rising from the rubble were groans, cries of despair from those buried alive*", Sandro Pertini, President of the Italian Republic, 25th November 1980 (two days after the earthquake). In the next section, an attempt to assess some of the most important factors is analyzed in depth.

### 4.2. Application of CEMs: Analysis of the Most Influential Factors

In addition to damage level, which is undoubtedly the most influential factor in the estimation of human casualties, building material and occupancy at the time of the event can significantly affect the results.

In order to analyze these two factors in depth and evaluate their influence on the final results, some comparisons, specifically based on the data of the 2009 L'Aquila earthquake considering Z&C and the NRA, were carried out. It is worth underlining that only the L'Aquila scenario was considered due to the homogeneous distribution of its building stock in terms of structural types, namely, URM (43%) and RC (40%) buildings. With regard to the selection of the CEMs to be compared, although based on a similar approach, Z&C and the NRA show two significant differences concerning the occupancy rate and the fatality rate. Indeed, unlike Z&C, in the NRA, the fatality rates are independent of building type, and the occupancy rate at the time of the event is not explicitly considered (see Table 6).

The two considered models were firstly compared in terms of the adopted fatality rate values (Figure 4a). Also, including the reduction related to the occupancy rate at the time of the event (72% obtained from C&S, as described above), the fatality rates of Z&C are higher than those adopted in the NRA, especially for RC buildings. For example, for the D5 damage level, the fatality rate value is equal to about 0.22 for Z&C, and 0.1 for the NRA. Figure 4b displays the number of casualties estimated for the L'Aquila scenario as a function of damage level and building type. In addition to the percentage of casualties compared to the number of inhabitants (along the abscissa secondary axis), Figure 4b shows (near each column) the percentage of casualties by damage level and building type compared to the total estimated by each CEM.

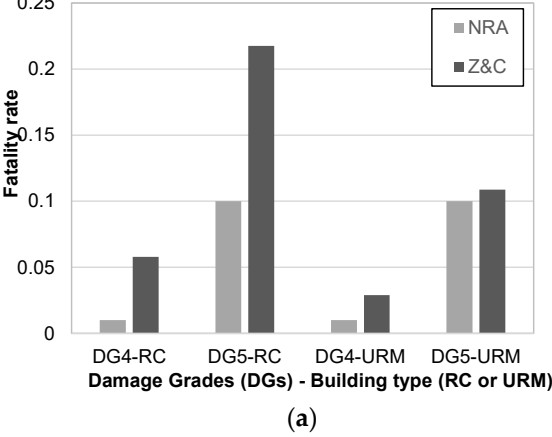

(**a**)

**Figure 4.** *Cont.*

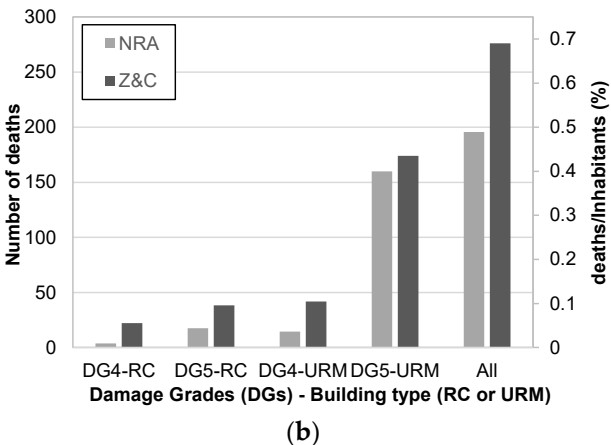

(**b**)

**Figure 4.** Fatality rates (**a**) and estimated number of casualties (**b**) as a function of damage level and building type related to the DB L'Aquila urban area for the NRA and Z&C models.

As expected, larger differences are found for RC buildings. Specifically, the percentage of total casualties due to RC buildings with D4 and D5 damage levels is equal to about 11% for the NRA, and about 22% for Z&C. A similar difference between the CEMs is observed for the number of casualties due to the partial collapse (i.e., D4) of masonry buildings.

In order to evaluate the influence of occupancy rate at the time of the earthquake on the final results, three different scenarios for L'Aquila were prepared using the Z&C model. Since Z&C does not provide any specific function to calculate the occupancy rate, the values from the curve proposed by Coburn and Spence [2] for residential buildings (and urban populations) were considered (see Figure 2):

(i)     Minimum (min) occupancy rate (i.e., daytime scenario) equal to 45%;
(ii)    Average (mean) occupancy rate equal to 65%;
(iii)   Maximum (max) occupancy rate (i.e., night-time scenario) equal to 78%.

Further, the "exact" value of the occupancy rate, i.e., corresponding to the time of the event (72%), was also considered.

First of all, the comparison in Figure 5 highlights that the occupancy rate values in Z&C produce significant differences, about +75% between the day- and night-time scenarios. Considering the actual number of deaths (equal to 220), Z&C overestimates the number of deaths in the cases of "mean" (+12%) and "max" (+35%) scenarios, while a lower value is found for the "min" scenario (−22%). The number of casualties obtained for the occupancy rate evaluated at the "exact" time of the event is about +25% greater than the actual deaths.

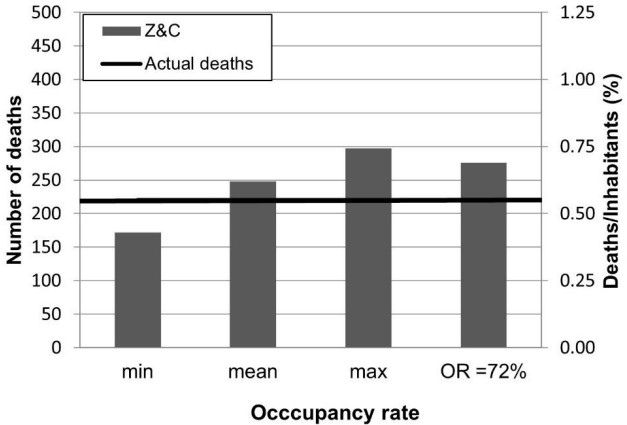

**Figure 5.** Comparison between the actual deaths from the urban area of L'Aquila and those obtained from Z&C by considering four different occupancy rate values (min = 45%, average = 65%, max = 68%, OR = 72%) as proposed by Coburn and Spence [2].

## 5. Occupancy Rate for Italian Residential Buildings

As a consequence of the role of the occupancy rate (OR) on expected casualties, and accounting for its inherent variability due to citizens' behavior, which can vary from country to country, it seems appropriate to specifically define the occupancy rate for the area under study [50]. Therefore, in the following, some analyses on the occupancy rates of residential buildings are carried out for the Italian context.

The data provided by the National Institute of Statistics (ISTAT [14]) related to "daily life and citizen opinions" were considered in terms of the typical occupancy of residential buildings during weekdays (related to a generic day) and holidays (associated with Sunday) for the age class "15 years and over". The data refer to four municipality classes in terms of number of inhabitants, that is, ≤2000; 2001–10,000; 10,001–50,000; and >50,000.

Figure 6 shows a comparison between the curves related to the typical occupancy of residential buildings on weekdays and holidays for the four classes of municipalities. In general, OR values are very different between the night-time interval (18:00–06:00) and the daytime interval (06:00–18:00). As for the night interval, OR increases from 18:00 to 00:00 and it is almost constant in the time period of 00:00–06:00, with a value of about 97%. As for the day interval, starting from 06:00, OR progressively reduces until 09:30–11:30 (this range is due to the different values obtained as a function of class of inhabitants and the day of week), when it reaches the minimum value (around 30–40%). After a "day peak" is reached at about 12:30–14:30 (when the OR value is about 60–65%), OR decreases again until 18:00, when it progressively increases up to the maximum "night" value.

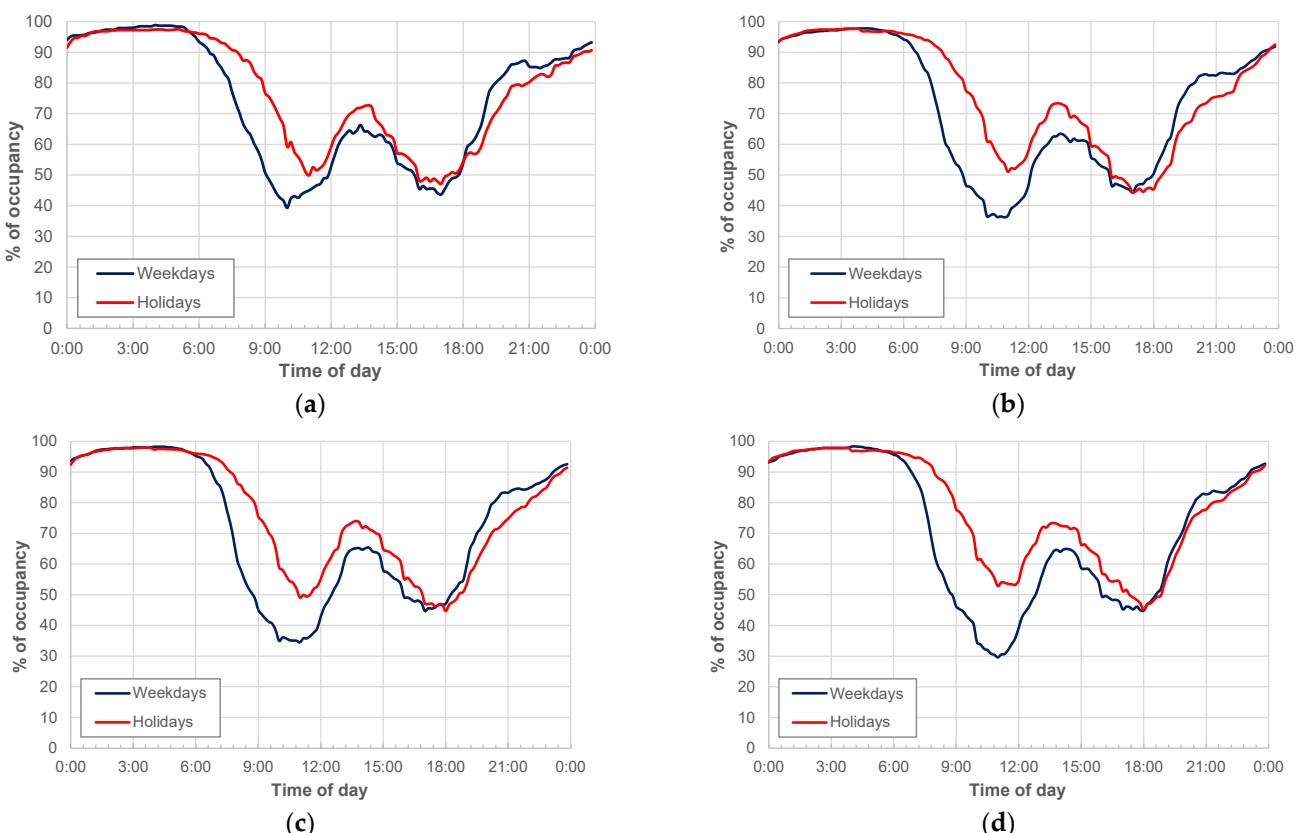

**Figure 6.** Comparison of the occupancy of residential buildings for weekdays (blue curve) and holidays (red curve) by considering the four classes of municipalities in terms of number of inhabitants: (**a**) ≤2000; (**b**) 2,001–10,000; (**c**) 10,001–50,000; and (**d**) >50,000.

Figure 6 highlights that OR related to holidays is generally higher than that of weekdays in the daytime interval, with lower differences in the night interval. The OR differences between weekday and holiday values increase with the size of the municipality, especially

in the daytime interval. In this regard, Figure 7 shows a comparison between the functions related to small (≤2000 inhabitants) and large (>50,000 inhabitants) municipalities for both weekdays and holidays. It mainly arises that the large municipalities have lower OR values in the daytime during weekdays compared to the smaller municipalities. Further, the "daytime peak" related to large municipalities is slightly delayed compared to small ones.

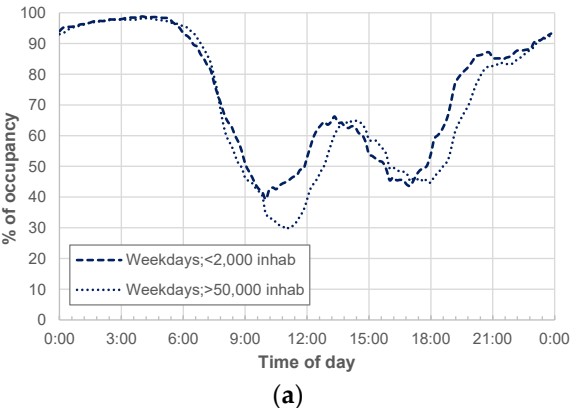 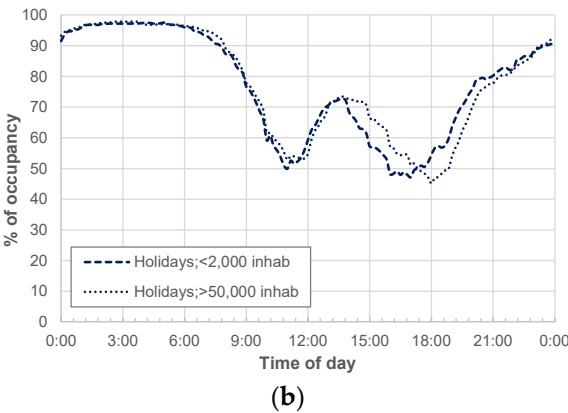

**(a)** **(b)**

**Figure 7.** Comparison of the occupancy of residential buildings between small (≤2000 inhabitants) and large (>50,000 inhabitants) municipalities for weekdays (**a**) and holidays (**b**).

Starting from the data reported above, Figure 8 shows the function for both "weekdays" (dashed blue line) and "holidays" (dashed red line) obtained via weighting over the number of municipalities, as classified in terms of number of inhabitants. For this purpose, Table 8 reports the number of municipalities related to each considered class (and the relative percentage) along with the number of inhabitants (and the relative percentage), as reported in the 2011 census of the Italian National Institute of Statistics [51].

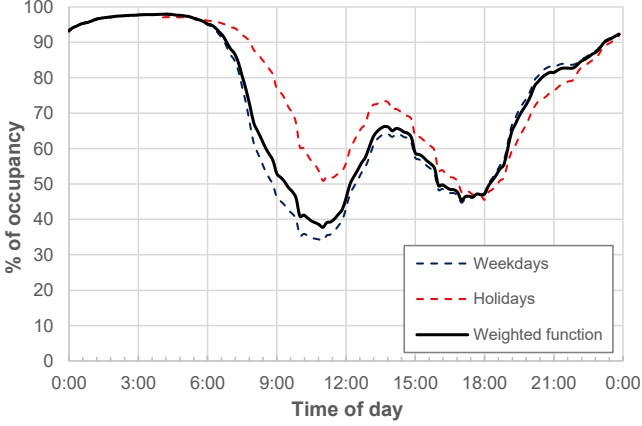

**Figure 8.** Occupancy rate function for weekdays (dashed blue line, only weighted over the municipalities size) and holidays (dashed red line, only weighted over the municipalities size). The solid black line refers to function weighted over the number of weekdays and holidays in one year.

In addition to the curves for weekdays and holidays, Figure 8 also shows the "weighted function" (solid black line) as obtained by considering the above-mentioned functions for weekdays and holidays, and then, additionally weighting them over the number of weekdays and holidays in one year.

In order to highlight the possible differences with the available OR functions, Figure 9 shows a comparison between the proposed functions and the ones provided by C&S for both urban and rural populations. Large differences arise from this comparison. As regards the "urban population" function, the maximum OR value is about 80%, which progressively reduces starting from midnight. On the contrary, according to the proposed function, the maximum value is about 95% and remains almost constant for a larger range of time,

i.e., from midnight to about 06:00. Although the minimum values are similar (in the range 40–45%) and occur approximately in the same time period (around the 9:00–12:00 interval), the proposed OR function shows two peak values, that is, about 65% at 14:00 (max) and about 45% at 17:00 (min), while C&S progressively increases up to the maximum value (at midnight).

**Table 8.** Distribution of Italian municipalities according to the 4 classes related to the number of inhabitants (first and second row) and distribution of the number of inhabitants for each class (third and fourth row).

|  | <2000 | 2000–10,000 | 10,000–50,000 | >50,000 | Total |
|---|---|---|---|---|---|
| # Municipalities | 3563 | 3326 | 1062 | 141 | 8092 |
| % Municipalities | 44% | 41% | 13% | 2% | -- |
| # Inhabitants in each class | 3,418,937 | 15,232,444 | 20,669,060 | 19,509,799 | 58,830,240 |
| % Inhabitants in each class | 6% | 26% | 35% | 34% | -- |

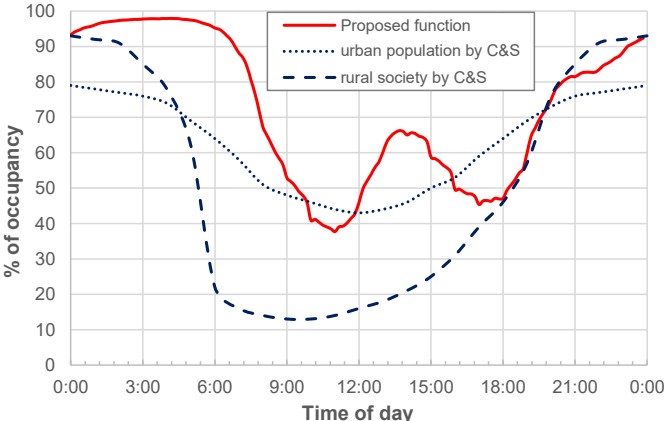

**Figure 9.** Comparison between the OR function as proposed by Coburn and Spence (2002) (for urban population, dotted blue line, and rural population, dashed blue line) and that proposed by the authors of the present paper.

By considering the "rural society" function, from 00:00 (at which the same OR value is found) until 18:00, it provides significantly lower values compared to the proposed function. For example, the minimum value is about 12% for C&S and about 40% for the proposed function. Such low values could be questionable, but what appears rather surprising, if not unconvincing, especially when compared to Italian habits, is the strong reduction in the OR found by C&S in the night-time interval starting from 00:00.

A further comparison that is useful to test the proposed OR function, specifically derived for Italy, compared to the one from C&S, whose data derive from different countries around the world, is reported in Figure 10. The estimated and actual numbers of deaths related to the two scenarios already described in Section 4, i.e., the 2009 L'Aquila and the 1980 Irpinia–Basilicata earthquakes, are compared. Regarding the estimated values, the two OR functions were, in turn, included in two "basic" models, the C&S and Z&C models, thus generating four combinations between the two "basic" models and the two OR functions.

With reference to the Irpinia–Basilicata scenario (an event that occurred at 19:34 local time), similar OR values can be obtained from the two considered functions, i.e., 67% for C&S and 69% for the OR proposed in this paper. As a consequence, for the same "basic" model, slight differences are found by varying the OR function. The estimated number of deaths obtained from the "basic" Z&C model including the proposed OR is 658 (−9% compared to the actual value), while it is 638 (−12%) considering the OR of C&S. On the

contrary, for the same OR, large differences arise from the two "basic" models. Specifically, C&S including the proposed OR provides 1358 deaths (+88% compared to the actual value), much higher than the value obtained considering the proposed OR in the Z&C model (658 deaths, −9% compared to the actual value).

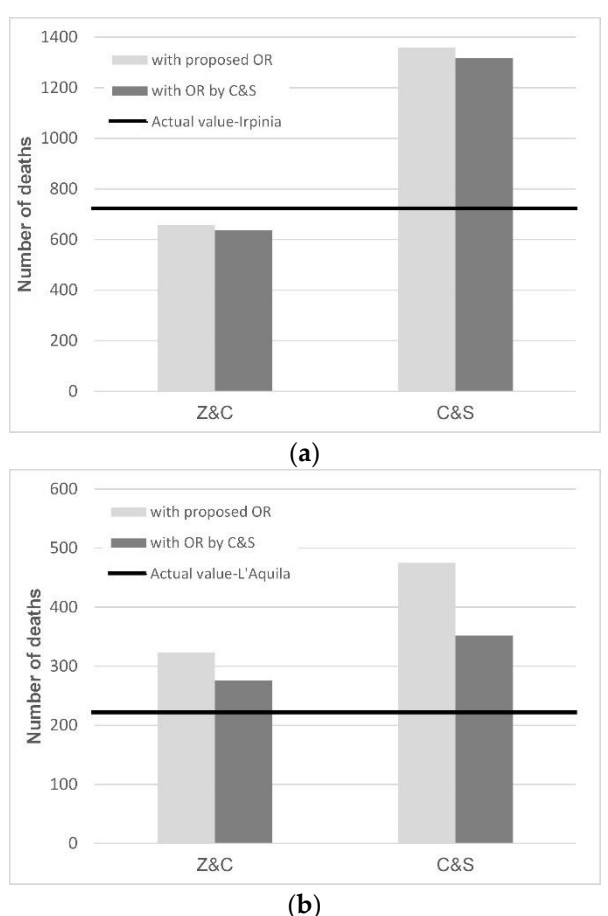

**Figure 10.** Number of deaths for the Irpinia–Basilicata (**a**) and L'Aquila (**b**) scenarios estimated using Z&C and C&S models including the proposed OR and OR of C&S, and comparison with the actual values.

As for the L'Aquila scenario (an event that occurred at 03:32 local time), the OR values are 97% and 72%, respectively, for the proposed OR function and the one of C&S. Of course, these large differences affect the number of estimated deaths that, for both "basic" models, overestimate the actual value. More specifically, when the "basic" Z&C model is considered, the number of deaths obtained including the proposed OR function is 323 (+47% over the actual value), while it is 276 (+25%) when considering the OR of C&S. Larger differences, both compared to the actual value and between the two considered OR functions, are found when adopting the "basic" C&S model. In total, there are 475 (+115%) and 352 (+60%) victims, respectively, when considering the proposed OR and that of C&S.

Due to the number of events considered, it is clear that the proposed OR function could not be verified on the basis of the above-described comparisons. However, some preliminary considerations can be made. In particular, the number of estimated casualties is found to be significantly dependent on the specific "basic" model considered compared to the adopted OR function: the differences computed using the same OR function applied to the two "basic" models are significantly larger than the ones found using the two OR functions with the same "basic" model. The C&S model, which was calibrated on the basis of a wide and less homogeneous database, provides a significant overestimation of the number of deaths, irrespective of the OR considered. On the contrary, the Z&C model,

whose data essentially refer to the Italian context, appears to be in good agreement with the actual values. In the case of the Irpinia–Basilicata scenario, the estimated value obtained with the Z&C model including the proposed OR function is very close to the actual value (the difference is about 9%).

A lower agreement (47%) is found in the case of L'Aquila, for which using the OR of C&S provides a better estimation compared to the actual value (25%). However, a remark needs to be made to better understand the overestimation found using the proposed OR: it is likely that the number of deaths caused by the L'Aquila event should not be referred to the maximum OR generally experienced at night time, i.e., when the earthquake occurred. Indeed, the main shock followed several foreshocks, which pushed some people to leave their dwellings, thus reducing the actual OR at the time of the event and, consequently, the actual number of victims.

## 6. Conclusions and Future Developments

In the definition of effective risk reduction plans, the estimation of expected losses due to future earthquakes is crucial. Further, seismic risk assessments for a given area in terms of expected human casualties, both deaths and injuries, can improve the preparedness of medical and relief agencies, thus contributing to effective emergency management.

As a matter of fact, estimating the number of casualties is a very complex task due to the numerous factors involved, including the large associated uncertainties, and the poor quality and scarcity of information generally available. Several studies and projects have been devoted in recent years to the development of Casualty Estimation Models (CEMs), but they have rarely been discussed and profitably compared, either with each other or with actual data from past earthquakes, through purposely performed loss scenarios.

In the present paper, an extensive literature overview of the main available CEMs was performed, and the most influential factors involved were analyzed. To figure out the prediction capability of the different CEMs, some comparative analyses were performed in terms of loss scenarios using data collected on two strong Italian earthquakes (i.e., 1980 Irpinia–Basilicata and 2009 L'Aquila). The comparison between the estimated and surveyed (actual) data brings out the contribution of some parameters accounted for in the considered CEMs. In addition to damage level, and specifically, collapse, which is found to be the most influential parameter, the consideration of building material and of building occupancy at the time of the event significantly impacts the estimation of human casualties.

On the basis of the data collected by the National Institute of Statistics related to "daily life and citizen opinions", the distributions of the occupancy rate (OR) for the Italian residential buildings were determined, considering weekdays and holidays and the size of the town under study (i.e., considering different classes of numbers of inhabitants). The occupancy rate functions were compared to the ones by Coburn and Spence [2], and their impact on casualty estimation was examined using the data from two scenarios of past Italian earthquakes (1980 Irpinia–Basilicata and 2009 L'Aquila).

The proposed OR functions are undoubtedly able to better represent the number of people living in Italian residential buildings during different hours of the day. However, the results from the comparisons, although promising, are not completely able to validate the proposed functions against the casualty numbers actually found in the considered events. Therefore, beyond additional investigations on the roles of the different factors involved in human casualty estimation, a CEM able to reliably consider the typical occupancy of the buildings at the "expected time of the event" and the role of the building material in the fatality rate need to be defined.

Further, since casualties due to the collapse of residential buildings constitute most, but not all, of the expected casualties, other possible causes need to be considered, including secondary catastrophes (e.g., landslides, rockfalls, urban fires), the collapse of large civil engineering structures, the direct effects of fault rupture and the unexpected collapse of public buildings.

　　　　Defining more reliable CEMs and bringing out the main factors determining casualties in cases of earthquakes, so that the driving causes can be progressively removed, appears more and more necessary. Updated data on the human impact of natural disasters (https://ourworldindata.org/natural-disasters, accessed on 6 July 202) shows that if, on one hand, "Historically, droughts and floods were the most fatal disaster events.", on the other hand, "Deaths from these events are now very low–the most deadly events today tend to be earthquakes."

**Author Contributions:** Conceptualization: V.M., G.N. and A.M.; methodology: V.M., G.N. and A.M.; validation: V.M., G.N. and A.M.; formal analysis: V.M., G.N. and A.M.; resources: V.M., G.N. and A.M.; data curation: V.M., G.N. and A.M.; writing—original draft preparation: V.M., G.N. and A.M.; writing—review and editing: V.M., G.N. and A.M.; visualization: V.M., G.N. and A.M.; supervision: V.M., G.N. and A.M. All authors have read and agreed to the published version of the manuscript.

**Funding:** This article was developed under the financial support of the Italian Department of Civil Protection, within the ReLUIS-DPC 2019–21 Research Project.

**Institutional Review Board Statement:** Not applicable.

**Informed Consent Statement:** Not applicable.

**Data Availability Statement:** References to datasets used in the analyses have been reported in the text.

**Conflicts of Interest:** The authors declare no conflict of interest.

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
