# Peer review of "Estimation of Human Casualties Due to Earthquakes: Overview and Application of Literature Models with Emphasis on Occupancy Rate"

_safety, 1980_

Round 1
Reviewer 1 Report
Comments and Suggestions for Authors
This article is a review/overview of several existing models in the literature regarding estimating victims due to earthquakes (Casualty Estimation Models - CEMs). Furthermore, the application covers two major earthquakes in Italy, Irpinia-Basilicata (1980) and L’Aquila (2009). Regarding the overview, several models in the literature are presented, as well as the different factors considered in each model. The influencing factors considered in each of these models (such as damage level, building classification, earthquake intensity, etc.) are not the same or are not considered in the same way in each of the models. In this way, the authors show that in addition to the level of damage, and specifically, the collapse, which is considered the most influential parameter, the consideration of the construction material and the occupancy of the building at the time of the event has a significant impact on the estimate of human casualties. The authors also make a different assessment of the occupancy rates (OR) of residential buildings in Italy, deriving specific distribution curves based on data from the National Institute of Statistics.
The manuscript has an interesting idea regarding comparing different CEMs and the results of these models when applied to actual earthquakes. However, concerning the results found when using the new occupancy rate curves proposed by the authors in the models, the comparisons' results are incapable of validating the proposed functions about the numbers of real victims found in the earthquakes considered.
In this way, the manuscript is interesting; however, I consider that many points need to be improved, including an overall review of the writing. Below are some examples of issues to correct in certain lines of the text.
Lines:
- 51: citation format for “Dolce et al. (2021)” is out of line with the rest.
207: should be “D5” and not “D4”.
- 291 and 292: problems with writing subscripts
325: the term “O_{RCI,D4/D5}” should be in parentheses.
459: the last 43% must be removed as it is repeated.
522-525: no justifications are provided for the chosen values of the parameters in this paragraph.
534: The value 490 appears to be incorrect. Would it be 290?
540: considering that the value for the Z&C model is greater than the actual value, wouldn't it be “overestimated” rather than “underestimated”?
Figure 4: The abbreviation “MUR” was used for Masonry. However, throughout the text the abbreviation “M” is used. The abbreviation must be standardized, and preferably without overlapping abbreviations, as “M” was also used for another variable in the manuscript.
Figure 5: this figure was not cited anywhere in the text. I believe it should have been cited in the paragraph ending on line 591.
598: the title of this section is repeated, and so is the numbering.
Figure 8: the figure should come before Table 8.
666: should be 18:00.
Figure 9: in figure 9, a figure identical to figure 8 was placed.
734: the paragraph begins with text without any meaning. “This section is not mandatory but can be added to the manuscript if the discussion is 734 unusually long or complex.”
- 789: reference number [2] is from 2002 and not 2022.
- several lines: the word “analyze” is used sometimes with “z” and sometimes with “s”.
In summary, although the present work did not find results of great relevance using the curve that the authors proposed for occupancy rates (OR), I think that if the work is improved at several points (such as those pointed out above), the This manuscript will have enough material to be published in Safety. Therefore, I can only recommend its publication AFTER the authors have taken into account the revisions and corrections.
Comments on the Quality of English LanguageModerate editing of English language is required
Reviewer 2 Report
Comments and Suggestions for Authors
There is no correspondence between Figure 5 and formulae (6) (7). The percentage of residence "OR" is smaller than "max" and gives a higher estimate of deaths. Please complete the records.
Please cite referenced papers in the same way (see e.g. line 51 or line 313).
Reviewer 3 Report
Comments and Suggestions for Authors
Present study focus on the examination of available models to estimate human casualties due to earthquakes. Useful comparisons are presented between the consider estimation models on actual loss, based on data from two earthquakes that happened in Italy. The main conclusion is that the occupancy rate functions play a key role to the estimation of human casualties and their rational examination in present work is very important.
Round 2
Reviewer 1 Report
Comments and Suggestions for Authors
I think the work has been improved in such a way that now the
manuscript has enough material to be published in Safety.
Reviewer 2 Report
Comments and Suggestions for Authors